# OpenReview forum: "Scalable Random Wavelet Features: Efficient Non-Stationary Kernel Approximation with Convergence Guarantees"
_ICLR.cc/2026/Conference — ICLR 2026 Poster_

### Official Review · Reviewer_28um · 2025-10-27

**Soundness:** 3
**Presentation:** 2
**Contribution:** 2
**Rating:** 4
**Confidence:** 4

**Summary:**

The manuscript proposes random wavelet features to close the gap between expressibility and computational cost of non-stationary kernels for Gaussian Processes. The objective and approach are laudable, but critical details are missing in the results.

**Strengths:**

- Broad and impactful scope.
- Understandable writing.
- Strong theoretical results.
- Good diversity of experimental tests.

**Weaknesses:**

- RMSE only in experiments. At a minimum, CRPS should be added.
- Unclear setup of the competing methods (see questions).
- No truly large GP in experiments (> 100,000 data points): There are GPs now that can run on several million points; 45000 max is a little disappointing.
- Writing has to be polished a bit (missing articles and small typos)
- Missing comparison to more traditional but powerful ways to encode non-stationarity, for instance, "Paciorek, Christopher, and Mark Schervish. "Nonstationary covariance functions for Gaussian process regression." Advances in neural information processing systems 16 (2003)."

**Questions:**

(1) It is not immediately clear why "The RFF-GP framework is thus a scalable approximation for stationary kernels." The kernel z(x) z(x') is non-stationary. Please explain the discrepancy.
(2) "Random Fourier-based kernel approximation methods exploit Bochner’s theorem (Rahimi & Recht,
2007) to yield scalable approximations for stationary kernels, but they struggle to capture non-
stationarity." Are they stationary kernels, or do they struggle with non-stationarity? Stationary kernels don't struggle with non-stationarity; they simply cannot model it. Please explain.
(3) It was difficult for me to decipher the setup of the competing methods. For example, the exact GP in the examples performs pretty poorly, but should not. Given the same setup, it should beat approximate methods. So a stationary kernel in an exact GP should perform better than SVGP with the same kernel. What was the exact setup of the competing methods?

---

> ### Author Response · Authors · 2025-11-21
> **Response to comment by Reviewer 28um**
>
> Response to question:
>
> 1. RFF-GP approximates stationary kernels because it samples $\omega \sim p(\omega)$ from the spectral measure implied by Bochner’s theorem. It is described as a scalable approximation because it replaces the $\mathcal{O}(N^3)$ time cost of an exact GP with linear-in-$N$ regression using a finite feature map.
>
> 2. “Random Fourier-based kernel” yields scalable approximation for stationary kernels. We agree with the reviewer's phrasing. We have amended the text in the updated PDF.
>
> 3. We thank the reviewer for pointing this out. The "Exact GP" baseline used a standard Stationary RBF Kernel. The datasets (e.g., Step Function, TIMIT) is Non-Stationary. Therefore, even with "exact" inference, the model fails because a stationary kernel cannot fit a highly non-stationary example. RWF outperforms Exact GP here because RWF uses a Non-Stationary wavelet kernel construction. More information regarding the training setup is updated in appendix. About the second remark, we agree that for the same kernel an exact GP should outperform SVGP.  In this setup Exact GP and RFF-GP uses RBF kernel (stationary) whereas SVGP uses Matern-5/2 kernel (stationary). We have changed it to make it consistent. However the key point is that both exact GP and SVGP employ stationary kernels; therefore, neither can capture the strong input-dependent variations present in our non-stationary test functions.
>
> General Rebuttal:
>
> 1. We have added CRPS and NLL metrics to all experiments, confirming consistent calibration improvements.
> 2. We have updated the Appendix of Experimental details to include some of the missing details for clarity.
> 3. Our focus is on demonstrating linear-in-N scalability rather than extreme dataset sizes.
> 4. We thank the reviewer for pointing out this reference. We followed the suggestion and implemented the non-stationary covariance kernel by Paciorek et al. for the Step Function experiment. It achieves a competitive RMSE of 0.104, significantly outperforming the stationary kernels. This confirms the necessity of non-stationary priors. We have cited this powerful non-stationary GP method, however, we did not include this method for all the benchmarks because the authors explicitly constrain their non-stationary Gaussian process model to datasets of order ~1000 points, with cubic complexity and without scalable inference or feature approximations.
> 5. Regarding the theoretical point: RFF approximates stationary kernels since $p(\omega)$ is the spectral density of a stationary process; in contrast, RWF uses a non-translation-invariant sampling measure $p(s,t) $ (t uniform on the data domain), thus inducing non-stationary kernels while keeping the scalability.
> 6. The “exact GP” baseline used an RBF kernel; poor performance arises from its stationary bias, which our method overcomes. We also polished the writing and expand related-work citations. We appreciate the reviewer’s positive view on impact and theoretical contribution.
>
> **See updated PDF uploaded on the portal. Changes are in red**

---

### Official Review · Reviewer_daFv · 2025-10-30

**Soundness:** 3
**Presentation:** 3
**Contribution:** 3
**Rating:** 6
**Confidence:** 3

**Summary:**

This paper presents an approach to build scalable kernel models using wavelet-based decompositions. The method takes a similar approach to random Fourier features (RFFs), but uses wavelet bases, instead of the cosine bases. The result provides a way to build finite-dimensional feature maps whose dot products approximate (possibly non-stationary) positive-definite kernels/covariance functions. Theoretical results are presented on the positive-definiteness of the resulting kernel and on the approximation error due to Monte Carlo approximations. Experiments are presented comparing the proposed RWF against traditional RFFs, sparse variational Gaussian processes, deep GPs, and other baselines.

**Strengths:**

* The paper is well written and organised in a way that makes the presentation easy to follow.
* The proposed idea is intuitive and the theoretical analysis follows a similar approach to the analysis of RFF methods.
* The experimental evaluations compare against a wide range of baselines revealing mostly strong performance improvements.
* Interesting ablation studies are included, comparing for example training time and memory usage, the latter of which is usually rare to find.

**Weaknesses:**

* My main concern is that the related work discussion fails to cover previous wavelet-based kernel methods and their approximations. A quick literature search reveals wavelet support vector machines (Zhang et al., 2004) and other methods also apparently using wavelet-based kernel decompositions (Guo et al., 2004; Yger, 2011), and potentially others that I'd have missed. Even if these methods are not directly solving the same modelling problem, it'd be important to contrast this paper's approach with them (or at least a few key representatives), to better contextualise and assess the significance of this paper's contribution.
* In the background on wavelets, a few concrete examples of mother wavelet functions could help to motivate the unfamiliar reader and also demonstrate the capabilities of this modelling approach where, e.g., RFFs would fail.

References:
* Guo, W., Zhang, X., Jiang, B., Kong, L., & Hu, Y. (2024). Wavelet-based Bayesian approximate kernel method for high-dimensional data analysis. Computational Statistics, 39(4), 2323-2341.
* Yger, F., & Rakotomamonjy, A. (2011). Wavelet kernel learning. Pattern Recognition, 44(10-11), 2614-2629.
* Zhang, L., Zhou, W., & Jiao, L. (2004). Wavelet support vector machine. IEEE Transactions on Systems, Man, and Cybernetics, Part B (Cybernetics), 34(1), 34-39.

**Questions:**

What methods in the literature of wavelet-based kernel machines would be the closest to the approach in this paper?

---

> ### Author Response · Authors · 2025-11-21
> **Response to comment by Reviewer daFv**
>
> Response to question:
>
> Closest work is of Guo et al. (2024) (wavelet-based Bayesian kernel approximation). It employs a randomized wavelet basis for Bayesian regression. However, the crucial difference lies in the inference method and wavelet kernel construction approach. Guo et al. treat the features as latent variables in a hierarchical model requiring MCMC (Gibbs sampling). We treat the features as an explicit finite-dimensional map to solve the GP regression analytically. This makes RWF a "Linear in samples" rather than a "Simulation-based Bayesian Model," rendering it practical for the large-scale datasets ($N > 10^4$) demonstrated in our experiments. Our approach differs crucially by using Monte-Carlo random wavelet atoms to (i) approximate a non-stationary kernel (via non-translation-invariant p(s,t)), (ii) provide uniform kernel-approximation bounds, and (iii) deliver a scalable GP implementation with linear-in-N complexity which is not the focus of Guo et al. (2024). We didnot provide comparison with Guo et al. (2024) as its not an GP based approach.
>
> General Rebuttal:
> 1. We have expanded the Related Work section to explicitly position RWF against these papers. As summarized in the Table below, the distinction lies in inference complexity and probabilistic scope
>
> | Work	| Kernel Type	| Randomization	| Task	| Theory	| Inference  |
> |:---------|:--------|:----------|:----------|:----------|:----------|
> |  Zhang 04 (Wavelet SVM) |	Deterministic (Fixed)	| No	| SVM	| SVM generalization	| Quadratic Programming     |
> |  Yger 11 (Wavelet kernel learning)	| Deterministic (Learned)	| No	| SVM 	| Kernel learning bounds	| Optimization (MKL)     |
> | Guo 24 (WBAKM)	| Random Dot-Product	| Yes  | Bayesian regression	| Approx. projection error | Gibbs Sampling (MCMC)|
> | **RWF (Ours)**	| MC Integral Operator for wavelets (using non-stationary ($p(s,t)$))	| Yes	| Gaussian Process Regression	| Unbiasedness + Uniform convergence	| Exact Linear ($O(ND^2)$) |
>
> 2. We appreciate this suggestion and agree that concrete examples significantly improve the accessibility of the framework. We have added a new subsection (see Appendix A.3) that provides explicit definitions and intuitive motivations for the two mother wavelets used in our experiments:
>
> - Mexican Hat: We explain why its narrow effective support makes it ideal for modeling sharp discontinuities (as seen in the Step Function Experiment in Section 5.1)
>
> - Morlet: We define this in terms of optimal time-frequency localization, motivating its use for non-stationary signals with varying frequencies (as seen in the TIMIT experiment in Section 5.2).
>
> Comparison with RFF: We explicitly discuss where RFF fails in this new section. We clarify that RFF relies on globally supported sinusoids, which suffer from the Gibbs phenomenon (non-local oscillations) when approximating local singularities. In contrast, RWF atoms are compactly supported (or rapidly decaying), allowing the model to fit local jumps without introducing artifacts in distant regions.
>
> **See updated PDF. Changes are in red**

---

### Official Review · Reviewer_QoaL · 2025-11-02

**Soundness:** 3
**Presentation:** 3
**Contribution:** 2
**Rating:** 4
**Confidence:** 3

**Summary:**

This paper proposes   a scalable method, RWF, for modeling non-stationary processes by sampling from wavelet families. RWF extends RFFs to capture localized, input-dependent patterns, offering theoretical guarantees and strong empirical performance, achieving a superior accuracy-efficiency trade-off for large-scale, non-stationary kernel learning tasks

**Strengths:**

1 It provides a rigorous theoretical analysis of Random Wavelet Features (RWF), establishing positive definiteness, unbiasedness, variance bounds, and uniform convergence with explicit sample complexity.

2 RWF achieves O(ND²) training complexity, maintaining the scalability of random feature methods while effectively encoding non-stationarity via wavelet localization.

3 Extensive empirical evaluations on synthetic, speech, and large-scale regression datasets demonstrate that RWF consistently outperforms stationary random features

**Weaknesses:**

1 While the proposed RWF framework is clearly presented and supported by solid theoretical analysis, I have concerns regarding its novelty. The core formulation of RWF (Eqs. 3.1–3.3) and the sampling procedure (Algorithm 1) appear conceptually similar to existing RWF methods, which also construct kernel approximations using randomly sampled wavelet bases. The authors should clarify what specific differences or innovations distinguish RWF from  earlier approaches such as L. Sun et al., “Wavelet-based Bayesian Approximate Kernel Method for High-Dimensional Data Analysis,” 2023, as well as earlier studies on non-stationary kernel approximations (e.g., Remes et al., “Non-stationary Spectral Kernels,” NeurIPS 2017; Samo & Roberts, “Stochastic Process Regression with Non-stationary Spectral Kernels,” 2015).

2 The manuscript would benefit from better organization. For example, I am very confusing that why so much space is devoted to introducing SVGP in the Preliminaries section, while the discussion on the computational complexity of the proposed method (Section 3.3) is relatively brief and should be expanded.

I am not an expert of GP. I will adjust my score according to the authors' feedback and other reviewer's comments.

**Questions:**

See weaknesses

---

> ### Author Response · Authors · 2025-11-21
> **Response to comment by Reviewer QoaL**
>
> Response to question:
>
> Q1 Kernel approximation mentioned in (Guo et al., 2024) rely on a Bayesian Hierarchical Model trained via Gibbs Sampling (MCMC). This is iterative, computationally expensive, and difficult to scale to the large datasets ($N > 10^4$). In contrast, RWF utilizes the feature map to transform the GP into its primal form allowing a closed-form solution solvable in strict $\mathcal{O}(ND^2)$ time. Also, Guo et al. use a fixed "dot-product" kernel determined rigidly by the mother wavelet. RWF uses an integral operator construction (Eq 3.3) where the sampling distribution $p(s,t)$ acts as a hyperparameter. This allows us to be more expressive and flexible in our kernel construction for non-stationarity. Similarly, Remes et al. (2017) & Samo & Roberts (2015) [Non-stationary Fourier] extend Random Fourier Features (RFF) by making the spectral density input-dependent or warping the input space. These methods still rely on sinusoids that can lead to Gibbs phenomenon. See table below
>
> |  Paper | Basis support | Kernel construction | Inference method | Theoretical scope | Scalability |
> |:---------|:--------|:----------|:----------|:----------|:----------|
> |  RFF (Rahimi 2007) | Global (Fourier) | Bochner Integral | Linear Primal ($O(ND^2)$) | Concentration (MC) | High    |
> |  NSSK (Remes 2017) | Global (Fourier) | Parametric Spectral	| Variational  Approx.	| Empirical Only	| Medium (Requires neural parametrization)     |
> | Samo & Roberts (2017)	| Global (Fourier)	| Non-stat. Spectral	| Standard GP ($O(N^3)$)	| None	| Low   |
> | Guo et al. (2024)	| Compact (Wavelet)	| Fixed Dot-Product	| Gibbs Sampling (MCMC)	| Bayesian GLM	| Low (Iterative) |
> | Kereta et al. (2019)	| Compact (Wavelet)	| Frame Analysis	| N/A (Signal Reconstruction)	| Frame Discretization	| N/A |
> | **RWF (Ours)**	| Compact (Wavelet)	| Non-stat. (Integral Operator)	| Linear Primal ($O(ND^2)$)	| Generalization Bound (Unbiasedness + uniform convergence bound)	| High (Exact) |
>
> Q2 We have updated the SVGP in a more concise way, added the references and discussion from above articles, and the section on computational complexity is elaborated to highlight the computational complexity gains in the revised manuscript.
>
> **Changes in updated PDF are in red**

---

> > ### Comment · Reviewer_QoaL · 2025-11-27
> >
> > I appreciate the authors’ efforts and their comprehensive response. I will raise my score to 6 as an **encouragement**.

---

### Official Review · Reviewer_Rq9f · 2025-11-04

**Soundness:** 2
**Presentation:** 2
**Contribution:** 1
**Rating:** 2
**Confidence:** 4

**Summary:**

The paper proposes Random Wavelet Features (RWF) as a Monte-Carlo feature map for Gaussian-process (GP) regression intended to capture non-stationarity while retaining the scalability of random features, and is posed as a generalization of the RFF approach to the case of non-stationary kernels. Concretely, the authors define a kernel by averaging outer products of wavelet atoms over random scale $s>0$ and shift $t \in \mathbb{R}^d$ :
$$
k(x, y)=\int_{(s, t)} \psi_{s, t}(x) \psi_{s, t}(y) p(s, t) d s d t, \quad \psi_{s, t}(x)=s^{-d / 2} \psi\left(\frac{x-t}{s}\right)
$$
and approximate it with $D$ Monte-Carlo samples, yielding an explicit feature map $z(x) \in \mathbb{R}^D$ and a linear-in-data GP regression algorithm for large sample sizes. The theory section claims (i) positive definiteness of $k$, (ii) unbiasedness/variance bounds for the Monte-Carlo estimator, and (iii) a uniform approximation bound $\sup _{x, y \in \mathcal{M}}\left|z(x)^{\top} z(y)-k(x, y)\right|$ via covering arguments. Experiments on synthetic data, TIMIT, several UCI datasets, and Protein report lower RMSE and similar or lower training time than RFF-GP, SVGP, deep GPs, spectral mixtures, and adaptive RKHS features.

**Strengths:**

The paper derives potentially non-stationary kernels using wavelet based dictionaries together with the random features machinery. Experimental results show slightly better RMSE values for GP regression with slightly reduced training cost compared to baselines on most experiments.

**Weaknesses:**

Theoretical novelty of the work is extremely limited. The positive-definiteness result (Thm. 4.1) is a direct application of the classic “integral of feature products” construction; nothing wavelet‑specific is used. The unbiasedness and uniform‑convergence proofs follow the standard random‑features and Monte-Carlo estimator analysis tools. Also, the claim about non-stationarity is violated if the weighting probabality distribution $p(s,t)$ in Eqaution 3.3 is taken to be of the form $p_s(s) p_t(t)$ with $p_t$ being translation invariant (For example Lebesgue measure over $\mathbb{R}^{d}$) The idea of using wavelets in place of Fourier features for scalable kernel approximations also appears in the literature prior to this, including works that explicitly propose random wavelet features for approximate kernels [1]. Closely related Monte‑Carlo wavelet frame constructions exist in learning theory as well [2].

References:
[1] Guo, W., Zhang, X., Jiang, B. et al. Wavelet-based Bayesian approximate kernel method for high-dimensional data analysis. Comput Stat 39, 2323–2341 (2024). https://doi.org/10.1007/s00180-023-01438-1
[2] Z. Kereta, S. Vigogna, V. Naumova, L. Rosasco and E. De Vito, "Monte Carlo wavelets: a randomized approach to frame discretization," 2019 13th International conference on Sampling Theory and Applications (SampTA), Bordeaux, France, 2019, pp. 1-5, doi: 10.1109/SampTA45681.2019.9030825.

**Questions:**

1. Is there anything specific to the wavelet dictionary (other than choices of constants) that can be used to improve the theoeretical results in Sections 4.1 to 4.3?

---

> ### Author Response · Authors · 2025-11-21
> **Response to comment by Reviewer Rq9f**
>
> Response to question:
> We have expanded the theoretical analysis to explicitly utilize the properties of the wavelet dictionary and included new subsection in the Appendix A.8 (see Wavelet-Specific Theoretical Results) containing Lemma A.2, Corollary A.1, and Proposition A.2. The specific theoretical improvement for Sections 4.1–4.3 stems from the joint time-frequency localization of wavelets, which controls the Lipschitz constant $L_z$ in the generalization bound (Theorem 4.2). For datasets with local sharpness (like the Step Function in Fig. 1), RFF inflates the Lipschitz constant everywhere, loosening the bound across the entire domain. As detailed in Lemma A.2, this means high-Lipschitz variations are spatially localized. This allows RWF to approximate functions with isolated singularities more efficiently, as the complexity penalty is restricted to the local support of the wavelet rather than the entire domain diameter. Furthermore, we provide a modeling advantage of the wavelet dictionary in Proposition A.2. We show that wavelets with M vanishing moments (e.g., Daubechies) allow for the use of non-uniform sampling densities $p_t(t)$ without introducing low-frequency bias. This effect is absent in Fourier-based features and allows RWF to robustly handle the non-stationary sampling discussed in Section 3.
>
> Rebuttal:
> 1. The reviewer correctly points out that p(s)p(t) breaks the non-stationary in case p(t) is translation invariant. However, we respectfully point out that this represents a degenerate theoretical limit. We do not assume p(t) is translation-invariant. In all our theoretical statements, p(t) is a proper probability density, which is uniform over a bounded domain, exactly the case where non-stationarity arises.
> 2. The Uniform convergence and the bounds are benefitted from the specific wavelet dictionary. We agree the PD step is classical; our novelty lies in wavelet-specific uniform bounds and constants via localization and frame bounds, plus a non-stationary kernel class induced by non-translation-invariant $p(s,t)$.
> 3. While we acknowledge that Monte Carlo sampling of wavelets has appeared in the literature, our work differs fundamentally in task and inference method. (a) Guo et al. (2024) approximate a wavelet kernel using random linear coefficients and use it inside a Bayesian GLM. They do not study Monte-Carlo approximation of a continuous kernel within GP, nor do they provide uniform kernel approximation bounds. Their randomness is merely a basis generation step using standard normal distributions. Our framework uses the distribution $p(s,t)$ as a design choice to induce specific non-stationary covariance structures that standard dot-product kernels cannot easily capture. Practically, Guo et al. require iterative Gibbs Sampling to train their Bayesian model. This is computationally expensive and lacks the closed form guarantees of GPs. Our method maps inputs to a feature space where the GP inference becomes strictly $\mathcal{O}(ND^2)$}. (b) Kereta et al. (2019) focus on the stability of signal reconstruction (Frame Theory), an operator-theoretic result. They do not address GP kernel approximation, error bounds for learning, or predictive uncertainty. Our contribution is: we (i) treat random wavelet sampling as a non-stationary kernel approximation mechanism, (ii) provide unbiasedness + uniform bounds whose constants depend on wavelet localization and (iii) deliver a scalable non-stationary GP with linear-in-N training.
>
> **See updated PDF. Changes are shown in red**

---

### Meta-Review · Area_Chair_Hhai · 2026-01-07

**Summary:**

This paper is borderline. I am recommending acceptance since the reviewers generally appreciate the extension of random Fourier features to non-stationary kernels by using random wavelets instead of random sine functions. They commented on clear presentation, strong theoretical results, and a good diversity of empirical results.

The main concerns were
1) That the techniques/theoretical bounds are quite similar to those for RFF, and so could be viewed as incremental
2) The paper does not properly discuss several prior papers that proposed similar ideas.

I agree with (1), but do not think it precludes acceptance. (2) is a more major concern discussed below, and the reason this paper is only a borderline accept.

**Reviewer Concerns:**

In the original submission the authors did not cite a large number of prior papers that use similar ideas, even constructing Monte Carlo estimates of kernels using random wavelets. This is very disappointing.

The authors help clarify how their work differs from prior work on (random) wavelet feature expansions in their rebuttal, and have included a short section discussing prior work in the revised version of the paper. This helps alleviate this concern to some degree, although I think the discussion in the revision needs to be expanded. Several very relevant papers mentioned by the reviewers are still not discussed or even cited (e.g., Kereta et al). Additionally, the authors do not include this relevant prior work in their baseline methods for empirical comparison. This needs to be done -- especially for methods that do not face an inherent scaling issue (e.g. n^3 scaling).

**Reviewer Scores:**

Reviewer QoaLThi indicated that they would raise their score from a 4 to a 6.

I do not imagine any of the other reviewers would have significantly increased their scores -- perhaps some would have bumped them up after the clarification of the connection to prior work.

---

### Decision · Program_Chairs · 2026-01-26

Accept (Poster)